# Adversarial vulnerability for any classifier

**Alhussein Fawzi**
DeepMind
afawzi@google.com

**Hamza Fawzi**
Department of Applied Mathematics
& Theoretical Physics
University of Cambridge
h.fawzi@damtp.cam.ac.uk

**Omar Fawzi**
ENS de Lyon[*]
omar.fawzi@ens-lyon.fr

## Abstract

Despite achieving impressive performance, state-of-the-art classifiers remain highly vulnerable to small, imperceptible, adversarial perturbations. This vulnerability has proven empirically to be very intricate to address. In this paper, we study the phenomenon of adversarial perturbations under the assumption that the data is generated with a smooth generative model. We derive fundamental upper bounds on the robustness to perturbations of any classification function, and prove the existence of adversarial perturbations that transfer well across different classifiers with small risk. Our analysis of the robustness also provides insights onto key properties of generative models, such as their smoothness and dimensionality of latent space. We conclude with numerical experimental results showing that our bounds provide informative baselines to the maximal achievable robustness on several datasets.

## 1   Introduction

Deep neural networks are powerful models that achieve state-of-the-art performance across several domains, such as bioinformatics [1, 2], speech [3], and computer vision [4, 5]. Though deep networks have exhibited very good performance in classification tasks, they have recently been shown to be unstable to adversarial perturbations of the data [6, 7]. In fact, very small and often imperceptible perturbations of the data samples are sufficient to fool state-of-the-art classifiers and result in incorrect classification. This discovery of the surprising vulnerability of classifiers to perturbations has led to a large body of work that attempts to design robust classifiers [8, 9, 10, 11, 12, 13]. However, advances in designing robust classifiers have been accompanied with stronger perturbation schemes that defeat such defenses [14, 15, 16].

In this paper, we assume that the data distribution is defined by a smooth generative model (mapping latent representations to images), and study theoretically the existence of small adversarial perturbations for arbitrary classifiers. We summarize our main contributions as follows:

- We show fundamental upper bounds on the robustness of any classifier to perturbations, which provides a baseline to the maximal achievable robustness. When the latent space of the data distribution is high dimensional, our analysis shows that *any* classifier is vulnerable to very small perturbations. Our results further suggest the existence of a tight relation between robustness and linearity of the classifier in the latent space.

- We prove the existence of adversarial perturbations that transfer across different classifiers. This provides theoretical justification to previous empirical findings that highlighted the existence of such transferable perturbations.

---

[*]Univ Lyon, ENS de Lyon, CNRS, UCBL, LIP, F-69342, Lyon Cedex 07, France

- We quantify the difference between the robustness to adversarial examples *in the data manifold* and *unconstrained* adversarial examples, and show that the two notions of robustness can be precisely related: for any classifier $f$ with in-distribution robustness $r$, there exists a classifier $\tilde{f}$ that achieves unconstrained robustness $r/2$. This further provides support to the empirical observations in [17, 18].

- We evaluate our bounds in several experimental setups (CIFAR-10 and SVHN), and show that they yield informative baselines to the maximal achievable robustness.

Our robustness analysis provides in turn insights onto desirable properties of generative models capturing real-world distributions. In particular, the intriguing generality of our analysis implies that when the data distribution is modeled through a smooth and generative model with high-dimensional latent space, there exist small-norm perturbations of images that fool humans for any discriminative task defined on the data distribution. If, on the other hand, it is the case that the human visual system is inherently robust to small perturbations (e.g., in $\ell_p$ norm), then our analysis shows that a distribution over natural images cannot be modeled by smooth and high-dimensional generative models. Going forward in modeling complex natural image distributions, our results hence suggest that low dimensional, non-smooth generative models are important constraints to capture the real-world distribution of images; not satisfying such constraints can lead to small adversarial perturbations for any classifier, including the human visual system.

## 2  Related work

It was proven in [19, 20] that for certain families of classifiers, there exist adversarial perturbations that cause misclassification of magnitude $O(1/\sqrt{d})$, where $d$ is the data dimension, provided the robustness to random noise is fixed (which is typically the case if e.g., the data is normalized). In addition, fundamental limits on the robustness of classifiers were derived in [19] for some simple classification families. Other works have instead studied the existence of adversarial perturbations, under strong assumptions on the data distribution [18, 21]. In this work, motivated by the success of generative models mapping latent representations with a normal prior, we instead study the existence of robust classifiers under this general data-generating procedure and derive bounds on the robustness that hold for any classification function. A large number of techniques have recently been proposed to improve the robustness of classifiers to perturbations, such as adversarial training [8], robust optimization [9, 10], regularization [11], distillation [12], stochastic networks [13], etc... Unfortunately, such techniques have been shown to fail whenever a more complex attack strategy is used [14, 15], or when it is evaluated on a more complex dataset. Other works have recently studied procedures and algorithms to provably guarantee a certain level of robustness [22, 23, 24, 25, 26], and have been applied to small datasets (e.g., MNIST). For large scale, high dimensional datasets, the problem of designing robust classifiers is entirely open. We finally note that adversarial examples for generative models have recently been considered in [27]; our aim here is however different as our goal is to bound the robustness of classifiers when data comes from a generative model.

## 3  Definitions and notations

Let $g$ be a generative model that maps latent vectors $z \in \mathcal{Z} := \mathbb{R}^d$ to the space of images $\mathcal{X} := \mathbb{R}^m$, with $m$ denoting the number of pixels. To generate an image according to the distribution of natural images $\mu$, we generate a random vector $z \sim \nu$ according to the standard Gaussian distribution $\nu = \mathcal{N}(0, I_d)$, and we apply the map $g$; the resulting image is then $g(z)$. This data-generating procedure is motivated by numerous previous works on generative models, whereby natural-looking images are obtained by transforming normal vectors through a deep neural network [28], [29], [30], [31], [32].[2] Let $f : \mathbb{R}^m \to \{1, \dots, K\}$ be a classifier mapping images in $\mathbb{R}^m$ to discrete labels $\{1, \dots, K\}$. The discriminator $f$ partitions $\mathcal{X}$ into $K$ sets $C_i = \{x \in \mathcal{X} : f(x) = i\}$ each of which corresponds to a different predicted label. The relative proportion of points in class $i$ is equal to $\mathbb{P}(C_i) = \nu(g^{-1}(C_i))$, the Gaussian measure of $g^{-1}(C_i)$ in $\mathcal{Z}$.

The goal of this paper is to study the *robustness* of $f$ to additive perturbations under the assumption that the data is generated according to $g$. We define two notions of robustness. These effectively measure the minimum distance one has to travel in image space to change the classification decision.

- **In-distribution robustness:** For $x = g(z)$, we define the in-distribution robustness $r_{\text{in}}(x)$ as follows:
$$r_{\text{in}}(x) = \min_{r \in \mathcal{Z}} \|g(z+r) - x\| \text{ s.t. } f(g(z+r)) \neq f(x),$$
where $\|\cdot\|$ denotes an arbitrary norm on $\mathcal{X}$. Note that the perturbed image, $g(z+r)$ is *constrained to lie in the image* of $g$, and hence belongs to the support of the distribution $\mu$.

- **Unconstrained robustness:** Unlike the in-distribution setting, we measure here the robustness to *arbitrary* perturbations in the image space; that is, the perturbed image is not constrained anymore to belong to the data distribution $\mu$.
$$r_{\text{unc}}(x) = \min_{r \in \mathcal{X}} \|r\| \text{ s.t. } f(x+r) \neq f(x).$$
This notion of robustness corresponds to the widely used definition of adversarial perturbations. It is easy to see that this robustness definition is smaller than the in-distribution robustness; i.e., $r_{\text{unc}}(x) \leq r_{\text{in}}(x)$.

In this paper, we assume that the generative model is smooth, in the sense that it satisfies a *modulus of continuity* property, defined as follows:

**Assumption 1.** *We assume that $g$ admits a monotone invertible modulus of continuity $\omega$; i.e.,*[3]

$$\forall z, z' \in \mathcal{Z}, \|g(z) - g(z')\| \leq \omega(\|z - z'\|_2). \tag{1}$$

Note that the above assumption is milder than assuming Lipschitz continuity. In fact, the Lipschitz property corresponds to choosing $\omega(t)$ to be a linear function of $t$. In particular, the above assumption does not require that $\omega(0) = 0$, which potentially allows us to model distributions with disconnected support.[4]

It should be noted that generator smoothness is a desirable property of generative models. This property is often illustrated empirically by generating images along a straight path in the latent space [30], and verifying that the images undergo gradual semantic changes between the two endpoints. In fact, smooth transitions is often used as a qualitative evidence that the generator has learned relevant factors of variation.

Fig. 1 summarizes the problem setting and notations. Assuming that the data is generated according to $g$, we analyze in the remainder of the paper the robustness of arbitrary classifiers to perturbations.

## 4 Analysis of the robustness to perturbations

### 4.1 Upper bounds on robustness

We state a general bound on the robustness to perturbations and derive two special cases to make more explicit the dependence on the distribution and number of classes.

**Theorem 1.** *Let $f : \mathbb{R}^m \to \{1, \ldots, K\}$ be an arbitrary classification function defined on the image space. Then, the fraction of datapoints having robustness less than $\eta$ satisfies:*

$$\mathbb{P}\left(r_{in}(x) \leq \eta\right) \geq \sum_{i=1}^{K} \left(\Phi(a_{\neq i} + \omega^{-1}(\eta)) - \Phi(a_{\neq i})\right), \tag{2}$$

*where $\Phi$ is the cdf of $\mathcal{N}(0, 1)$, and $a_{\neq i} = \Phi^{-1}\left(\mathbb{P}\left(\bigcup_{j \neq i} C_j\right)\right)$.*

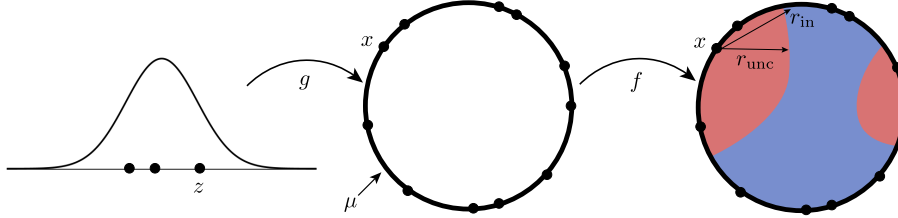

Figure 1: Setting used in this paper. The data distribution is obtained by mapping $\mathcal{N}(0, I_d)$ through $g$ (we set $d = 1$ and $g(z) = (\cos(2\pi z), \sin(2\pi z))$ in this example). The thick circle indicates the support of the data distribution $\mu$ in $\mathbb{R}^m$ ($m = 2$ here). The binary discriminative function $f$ separates the data space into two classification regions (red and blue colors). While the in-distribution perturbed image is required to belong to the data support, this is not necessarily the case in the unconstrained setting. In this paper, we do not put any assumption on $f$, resulting in potentially arbitrary partitioning of the data space. While the existence of very small adversarial perturbations seems counter-intuitive in this low-dimensional illustrative example (i.e., $r_{\text{in}}$ and $r_{\text{unc}}$ can be large for some choices of $f$), we show in the next sections that this is the case in high dimensions.

*In particular, if for all $i$, $\mathbb{P}(C_i) \leq \frac{1}{2}$ (the classes are not too unbalanced), we have*

$$\mathbb{P}\left(r_{in}(x) \leq \eta\right) \geq 1 - \sqrt{\frac{\pi}{2}} e^{-\omega^{-1}(\eta)^2/2} . \tag{3}$$

*To see the dependence on the number of classes more explicitly, consider the setting where the classes are equiprobable, i.e., $\mathbb{P}(C_i) = \frac{1}{K}$ for all $i$, $K \geq 5$, then*

$$\mathbb{P}\left(r_{in}(x) \leq \eta\right) \geq 1 - \sqrt{\frac{\pi}{2}} e^{-\omega^{-1}(\eta)^2/2} e^{-\eta\sqrt{\log\left(\frac{K^2}{4\pi \log(K)}\right)}} . \tag{4}$$

This theorem is a consequence of the Gaussian isoperimetric inequality first proved in [33] and [34]. The proofs can be found in the supplementary material.

**Remark 1. Interpretation.** For easiness of interpretation, we assume that the function $g$ is Lipschitz continuous, in which case $\omega^{-1}(\eta)$ is replaced with $\eta/L$ where $L$ is the Lipschitz constant. Then, Eq. (3) shows the existence of perturbations of norm $\eta \propto L$ that can fool any classifier. This norm should be compared to the typical norm given by $\mathbb{E}\|g(z)\|$. By normalizing the data, we can assume $\mathbb{E}\|g(z)\| = \mathbb{E}\|z\|_2$ without loss of generality.[5] As $z$ has a normal distribution, we have $\mathbb{E}\|z\|_2 \in [\sqrt{d-1}, \sqrt{d}]$ and thus the typical norm of an element in the data set satisfies $\mathbb{E}\|g(z)\| \geq \sqrt{d-1}$. Now if we plug in $\eta = 2L$, we obtain that the robustness is less than $2L$ with probability exceeding 0.8. This should be compared to the typical norm which is at least $\sqrt{d-1}$. Our result therefore shows that when $d$ is large and $g$ is smooth (in the sense that $L \ll \sqrt{d}$), there exist small adversarial perturbations that can fool arbitrary classifiers $f$. Fig. 2 provides an illustration of the upper bound, in the case where $\omega$ is the identity function.

**Remark 2. Dependence on $K$.** Theorem 1 shows an *increasing* probability of misclassification with the number of classes $K$. In other words, it is easier to find adversarial perturbations in the setting where the number of classes is large, than for a binary classification task.[6] This dependence confirms empirical results whereby the robustness is observed to decrease with the number of classes. The dependence on $K$ captured in our bounds is in contrast to previous bounds that showed decreasing probability of fooling the classifier, for larger number of classes [20].

**Remark 3. Classification-agnostic bound.** Our bounds hold for any classification function $f$, and are not specific to a family of classifiers. This is unlike the work of [19] that establishes bounds on the robustness for specific classes of functions (e.g., linear or quadratic classifiers).

**Remark 4. How tight is the upper bound on robustness in Theorem 1?** Assuming that the smoothness assumption in Eq. 1 is an equality, let the classifier $f$ be such that $f \circ g$ separates the latent space into $B_1 = g^{-1}(C_1) = \{z : z_1 \geq 0\}$ and $B_2 = g^{-1}(C_2) = \{z : z_1 < 0\}$. Then, it follows that

$$\begin{aligned}
\mathbb{P}(r_{\mathrm{in}}(x)) \leq \eta) &= \mathbb{P}(\exists r : \|g(z+r) - g(z)\| \leq \eta, f(g(z+r)) \neq f(g(z)))) \\
&= \mathbb{P}(\exists r : \|r\|_2 \leq \omega^{-1}(\eta), \mathrm{sgn}(z_1 + r_1)\mathrm{sgn}(z_1) < 0) \\
&= \mathbb{P}(z \in B_1, z_1 < \omega^{-1}(\eta)) + \mathbb{P}(z \in B_2, z_1 \geq -\omega^{-1}(\eta)) = 2(\Phi(\omega^{-1}(\eta)) - \Phi(0)),
\end{aligned}$$

which precisely corresponds to Eq. (2). In this case, the bound in Eq. (2) is therefore an equality. More generally, this bound is an equality if the classifier induces linearly separable regions in the latent space.[7] This suggests that classifiers are maximally robust when the induced classification boundaries in the latent space are linear. We stress on the fact that boundaries in the $\mathcal{Z}$-space can be very different from the boundaries in the image space. In particular, as $g$ is in general non-linear, $f$ might be a highly *non-linear function* of the input space, while $z \mapsto (f \circ g)(z)$ is a linear function in $z$. We provide an explicit example in the supplementary material illustrating this remark.

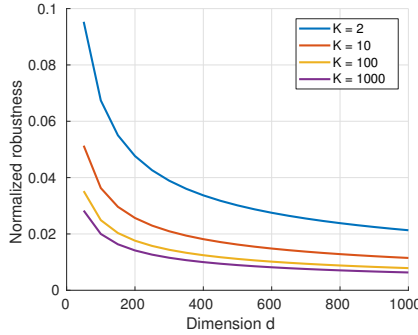

Figure 2: Upper bound (Theorem 1) on the median of the normalized robustness $r_{\mathrm{in}}/\sqrt{d}$ for different values of the number of classes $K$, in the setting where $\omega(t) = t$. We assume that classes have equal measure (i.e., $\mathbb{P}(C_i) = 1/K$).

**Remark 5. Adversarial perturbations in the latent space** While the quantities introduced in Section 3 measure the robustness in the *image space*, an alternative is to measure the robustness in the *latent space*, defined as $r_Z = \min_r \|r\|_2$ s.t. $f(g(z+r)) \neq f(g(z))$. For natural images, latent vectors provide a decomposition of images into meaningful factors of variation, such as features of objects in the image, illumination, etc... Hence, perturbations of vectors in the latent space measure the amount of change one needs to apply to such meaningful latent features to cause data misclassification. A bound on the magnitude of the minimal perturbation in the latent space (i.e., $r_Z$) can be directly obtained from Theorem 1 by setting $\omega$ to identity (i.e., $\omega(t) = t$). Importantly, note that no assumptions on the smoothness of the generator $g$ are required for our bounds to hold when considering this notion of robustness.

**Relation between in-distribution robustness and unconstrained robustness.**

While the previous bound is specifically looking at the in-distribution robustness, in many cases, we are interested in achieving *unconstrained* robustness; that is, the perturbed image is not constrained to belong to the data distribution (or equivalently to the range of $g$). It is easy to see that any bound derived for the in-distribution robustness $r_{\mathrm{in}}(x)$ also holds for the unconstrained robustness $r_{\mathrm{unc}}(x)$ since it clearly holds that $r_{\mathrm{unc}}(x) \leq r_{\mathrm{in}}(x)$. One may wonder whether it is possible to get a better upper bound on $r_{\mathrm{unc}}(x)$ directly. We show here that this is not possible if we require our bound to hold for any general classifier. Specifically, we construct a family of classifiers for which $r_{\mathrm{unc}}(x) \geq \frac{1}{2}r_{\mathrm{in}}(x)$, which we now present:

For a given classifier $f$ in the image space, define the classifier $\tilde{f}$ constructed in a nearest neighbour strategy:

$$\tilde{f}(x) = f(g(z^*)) \quad \text{with} \quad z^* = \arg\min_z \|g(z) - x\|. \tag{5}$$

Note that $\tilde{f}$ behaves exactly in the same way as $f$ on the image of $g$ (in particular, it has the same risk and in-distribution robustness). We show here that it has an unconstrained robustness that is at least half of the in-distribution robustness of $f$.

**Theorem 2.** *For the classifier $\tilde{f}$, we have $r_{unc}(x) \geq \frac{1}{2}r_{in}(x)$.*

This result shows that if a classifier has in-distribution robustness $r$, then we can construct a classifier with unconstrained robustness $r/2$, through a simple modification of the original classifier $f$. Hence, classification-agnostic limits derived for both notions of robustness are essentially the same. It should further be noted that the procedure in Eq. (5) provides a constructive method to increase the robustness of any classifier to unconstrained perturbations. Such a nearest neighbour strategy is useful when the in-distribution robustness is much larger than the unconstrained robustness, and permits the latter to match the former. This approach has recently been found to be successful in increasing the robustness of classifiers when accurate generative models can be learned in [35]. Other techniques [17] build on this approach, and further use methods to increase the in-distribution robustness.

## 4.2 Transferability of perturbations

One of the most intriguing properties about adversarial perturbations is their transferability [6, 36] across different models. Under our data model distribution, we study the existence of transferable adversarial perturbations, and show that two models with approximately zero risk will have shared adversarial perturbations.

**Theorem 3** (Transferability of perturbations). *Let $f, h$ be two classifiers. Assume that $\mathbb{P}(f \circ g(z) \neq h \circ g(z)) \leq \delta$ (e.g., if $f$ and $h$ have a risk bounded by $\delta/2$ for the data set generated by $g$). In addition, assume that $\mathbb{P}(C_i(f)) + \delta \leq \frac{1}{2}$ for all $i$.[8] Then,*

$$\mathbb{P}\left\{\exists v : \|v\|_2 \leq \eta \text{ and } \begin{array}{l} f(g(z) + v) \neq f(g(z)) \\ h(g(z) + v) \neq h(g(z)) \end{array}\right\}$$
$$\geq 1 - \sqrt{\frac{\pi}{2}}e^{-\omega^{-1}(\eta)^2/2} - 2\delta. \tag{6}$$

Compared to Theorem 1 which bounds the robustness to adversarial perturbations, the extra price to pay here to find *transferable* adversarial perturbations is the $2\delta$ term, which is small if the risk of both classifiers is small. Hence, our bounds provide a theoretical explanation for the existence of transferable adversarial perturbations, which were previously shown to exist in [6, 36]. The existence of transferable adversarial perturbations across several models with small risk has important security implications, as adversaries can, in principle, fool different classifiers with a single, classifier-agnostic, perturbation. The existence of such perturbations significantly reduces the difficulty of attacking (potentially black box) machine learning models.

## 4.3 Approximate generative model

In the previous results, we have assumed that the data distribution is exactly described by the generative model $g$ (i.e., $\mu = g_*(\nu)$ where $g_*(\nu)$ is the pushforward of $\nu$ via $g$). However, in many cases, such generative models only provide an *approximation* to the true data distribution $\mu$. In this section, we specifically assume that the generated distribution $g_*(\nu)$ provides an approximation to the true underlying distribution in the 1-Wasserstein sense on the metric space $(\mathcal{X}, \|\cdot\|)$; i.e., $W(g_*(\nu), \mu) \leq \delta$, and derive upper bounds on the robustness. This assumption is in line with recent advances in generative models, whereby the generator provides a good approximation (in the Wasserstein sense) to the true distribution, but does not exactly fit it [31]. We show here that similar upper bounds on the robustness (in expectation) hold, as long as $g_*(\nu)$ provides an accurate approximation of the true distribution $\mu$.

**Theorem 4.** *We use the same notations as in Theorem 1. Assume that the generator $g$ provides a $\delta$ approximation of the true distribution $\mu$ in the 1-Wasserstein sense on the metric space $(\mathcal{X}, \| \cdot \|)$; that is, $W(g_*(\nu), \mu) \leq \delta$ (where $g_*(\nu)$ is the pushforward of $\nu$ via $g$), the following inequality holds provided $\omega$ is concave*

$$\mathbb{E}_{x \sim \mu} r_{unc}(x) \leq \omega \left( \sum_{i=1}^{K} -a_{\neq i} \Phi(-a_{\neq i}) + \frac{e^{-a_{\neq i}^2/2}}{\sqrt{2\pi}} \right) + \delta,$$

*where $r_{unc}(x)$ is the unconstrained robustness in the image space. In particular, for $K \geq 5$ equiprobable classes, we have*

$$\mathbb{E}_{x \sim \mu} r_{unc}(x) \leq \omega \left( \frac{\log(4\pi \log(K))}{\sqrt{2 \log(K)}} \right) + \delta.$$

In words, when the data is defined according to a distribution which can be *approximated* by a smooth, high-dimensional generative model, our results show that arbitrary classifiers will have small adversarial examples in expectation. We also note that as $K$ grows, this bound decreases and even goes to zero under the sole condition that $\omega$ is continuous at 0. Note however that the decrease is slow as it is only logarithmic.

## 5 Experimental evaluation

We now evaluate our bounds on the SVHN dataset [37] which contains color images of house numbers, and the task is to classify the digit at the center of the image. In all this section, computations of perturbations are done using the algorithm in [38].[9] The dataset contains $73,257$ training images, and $26,032$ test images (we do not use the images in the 'extra' set). We train a DCGAN [30] generative model on this dataset, with a latent vector dimension $d = 100$, and further consider several neural networks architectures for classification.[10] For each classifier, the empirical robustness is compared to our upper bound.[11] In addition to reporting the in-distribution and unconstrained robustness, we also report the robustness in the latent space: $r_Z = \min_r \|r\|_2$ s.t. $f(g(z + r)) \neq f(g(z))$. For this robustness setting, note that the upper bound exactly corresponds to Theorem 1 with $\omega$ set to the identity map. Results are reported in Table 1.

Observe first that the upper bound on the robustness in the latent space is of the same order of magnitude as the empirical robustness computed in the $\mathcal{Z}$-space, for the different tested classifiers. This suggests that the isoperimetric inequality (which is the only source of inequality in our bound, when factoring out smoothness) provides a reasonable baseline that is on par with the robustness of best classifiers. In the image space, the theoretical prediction from our classifier-agnostic bounds is one order of magnitude larger than the empirical estimates. Note however that our bound is still non-vacuous, as it predicts the norm of the required perturbation to be approximately $1/3$ of the norm of images (i.e., normalized robustness of $0.36$). This potentially leaves room for improving the robustness in the image space. Moreover, we believe that the bound on the robustness in the image space is not tight (unlike the bound in the $\mathcal{Z}$ space) as the smoothness assumption on $g$ can be conservative.

Further comparisons of the figures between in-distribution and unconstrained robustness in the image space interestingly show that for the simple LeNet architecture, a large gap exists between these two quantities. However, by using more complex classifiers (ResNet-18 and ResNet-101), the gap between in-distribution and unconstrained robustness gets smaller. Recall that Theorem 2 says that any classifier can be modified in a way that the in-distribution robustness and unconstrained robustness

|  | Upper bound on robustness | 2-Layer LeNet | ResNet-18 | ResNet-101 |
|---|---|---|---|---|
| Error rate | - | 11% | 4.8% | 4.2 % |
| Robustness in the $\mathcal{Z}$-space | $16 \times 10^{-3}$ | $6.1 \times 10^{-3}$ | $6.1 \times 10^{-3}$ | $6.6 \times 10^{-3}$ |
| In-distribution robustness | $36 \times 10^{-2}$ | $3.3 \times 10^{-2}$ | $3.1 \times 10^{-2}$ | $3.1 \times 10^{-2}$ |
| Unconstrained robustness | $36 \times 10^{-2}$ | $0.39 \times 10^{-2}$ | $1.1 \times 10^{-2}$ | $1.4 \times 10^{-2}$ |

Table 1: Experiments on SVHN dataset. We report the $25\%$ percentile of the *normalized* robustness at each cell, where probabilities are computed either theoretically (for the upper bound) or empirically. More precisely, we report the following quantities for the upper bound column: For the **robustness in the $\mathcal{Z}$ space**, we report $t/\mathbb{E}(\|z\|_2)$ such that $\mathbb{P}\left(\min_r \|r\|_2 \text{ s.t. } f(g(z+r)) \neq f(g(z)) \leq t\right) \geq 0.25$, using Theorem 1 with $\omega$ taken as identity. For the **robustness in image-space**, we report $t/\mathbb{E}(\|g(z)\|_2)$ such that $\mathbb{P}\left(r_{\text{in}}(x) \leq t\right) \geq 0.25$, using Theorem 1, with $\omega$ estimated empirically (Section C.2 in supp. material).

|  | Upper bound on robustness | VGG [40] | Wide ResNet [41] | Wide ResNet + Adv. training [10, 15] |
|---|---|---|---|---|
| Error rate | - | 5.5% | 3.9% | 16.0% |
| Robustness in the $\mathcal{Z}$-space | 0.016 | $2.5 \times 10^{-3}$ | $3.0 \times 10^{-3}$ | $3.6 \times 10^{-3}$ |
| In-distribution robustness | 0.10 | $4.8 \times 10^{-3}$ | $5.9 \times 10^{-3}$ | $8.3 \times 10^{-3}$ |
| Unconstrained robustness | 0.10 | $0.23 \times 10^{-3}$ | $0.20 \times 10^{-3}$ | $2.0 \times 10^{-3}$ |

Table 2: Experiments on CIFAR-10 (same setting as in Table 1). See supp. for details about models.

only differ by a factor 2, while preserving the accuracy. But this modification may result in a more complicated classifier compared to the original one; for example starting with a linear classifier, the modified classifier will in general not be linear. This interestingly matches with our numerical values for this experiment, as the multiplicative gap between in-distribution and unconstrained robustness approaches 2 as we make the classification function more complex (e.g., in-distribution robustness of $3.1 \times 10^{-2}$ and out-distribution $1.4 \times 10^{-2}$ for ResNet-101).

We now consider the more complex CIFAR-10 dataset [39]. The CIFAR-10 dataset consists of 10 classes of $32 \times 32$ color natural images. Similarly to the previous experiment, we used a DCGAN generative model with $d = 100$, and tested the robustness of state-of-the-art deep neural network classifiers. Quantitative results are reported in Table 2. Our bounds notably predict that any classifier defined on this task will have perturbations not exceeding $1/10$ of the norm of the image, for $25\%$ of the datapoints in the distribution. Note that using the PGD adversarial training strategy of [10] (which constitutes one of the most robust models to date [15]), the robustness is significantly improved, despite still being $\sim 1$ order of magnitude smaller than the baseline of $0.1$ for the in-distribution robustness. The construction of more robust classifiers, alongside better empirical estimates of the quantities involved in the bound/improved bounds will hopefully lead to a convergence of these two quantities, hence guaranteeing optimality of the robustness of our classifiers.

# 6 Discussion

We have shown the existence of a baseline robustness that no classifier can surpass, whenever the distribution is approximable by a generative model mapping latent representations to images. The bounds lead to informative numerical results: for example, on the CIFAR-10 task (with a DCGAN approximator), our upper bound shows that a significant portion of datapoints can be fooled with a perturbation of magnitude $10\%$ that of an image. Existing classifiers however do not match the derived upper bound. Moving forward, we expect the design of more robust classifiers to get closer to this upper bound. The existence of a baseline robustness is fundamental in that context in order to measure the progress made and compare to the optimal robustness we can hope to achieve.

In addition to providing a baseline, this work has several practical implications on the *robustness* front. To construct classifiers with better robustness, our analysis suggests that these should have linear decision boundaries in the *latent* space; in particular, classifiers with multiple disconnected classification regions will be more prone to small perturbations. We further provided a constructive way to provably close the gap between unconstrained robustness and in-distribution robustness.

Our analysis at the intersection of classifiers' robustness and generative modeling has further led to insights onto *generative models*, due to its intriguing generality. If we take as a premise that human visual system classifiers require large-norm perturbations to be fooled (which is implicitly assumed in many works on adversarial robustness, though see [42]), our work shows that natural image distributions cannot be modeled as very high dimensional and smooth mappings. While current dimensions used for the latent space (e.g., $d = 100$) do not lead to any contradiction with this assumption (as upper bounds are sufficiently large), moving to higher dimensions for more complex datasets might lead to very small bounds. To model such datasets, the prior distribution, smoothness and dimension properties should therefore be carefully set to avoid contradictions with the premise. For example, conditional generative models can be seen as non-smooth generative models, as different generating functions are used for each class. We finally note that the derived results do bound the *norm* of the perturbation, and not the human perceptibility, which is much harder to quantify. We leave it as an open question to derive bounds on more perceptual metrics.

### Acknowledgments

A.F. would like thank Seyed Moosavi, Wojtek Czarnecki, Neil Rabinowitz, Bernardino Romera-Paredes and the DeepMind team for useful feedbacks and discussions.

## Footnotes

[2]Instead of sampling from $\mathcal{N}(0, I_d)$ in $\mathcal{Z}$, some generative models sample from the uniform distribution in $[-1, 1]^d$. The results of this paper can be easily extended to such generative procedures.

[3]This assumption can be extended to random $z$ (see C.2 in the supp. material). For ease of exposition however, we use here the deterministic assumption.

[4]In this paper, we use the term *smooth* generative models to denote that the function $\omega(\delta)$ takes small values for small $\delta$.

[5]Without this assumption, the following discussion applies if we replace the Lipschitz constant with the normalized Lipschitz constant $L' = L \frac{\mathbb{E}\|z\|_2}{\mathbb{E}\|g(z)\|}$.

[6]We assume here equiprobable classes.

[7]In the case where Eq. (1) is an inequality, we will not exactly achieve the bound, but get closer to it when $f \circ g$ is linear.

[8]This assumption is only to simplify the statement, a general statement can be easily derived in the same way.

[9]Note that in order to estimate robustness quantities (e.g., $r_{\text{in}}$), we do not need the ground truth label, as the definition only involves the change of the estimated label. Estimation of the robustness can therefore be readily done for automatically generated images.

[10]For the SVHN and CIFAR-10 experiments, we show examples of generated images and perturbed images in the supplementary document (Section C.3). Moreover, we provide in C.1 details on the architectures of the used models.

[11]To evaluate numerically the upper bound, we have used a probabilistic version of the modulus of continuity, where the property is not required to be satisfied for *all* $z, z'$, but rather with high probability, and accounted for the error probability in the bound. We refer to the supp. material for the detailed optimization used to estimate the smoothness parameters.

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
