[Supplementary Material · Adversarial_vulnerability_for_any_classifier_NEURIPS_2018_supplementary.pdf]

# Adversarial vulnerability for any classifier
# (Supplementary material)

**Alhussein Fawzi**
DeepMind
afawzi@google.com

**Hamza Fawzi**
Department of Applied Mathematics
& Theoretical Physics
University of Cambridge
h.fawzi@damtp.cam.ac.uk

**Omar Fawzi**
ENS de Lyon*
omar.fawzi@ens-lyon.fr

## A   Proofs

### A.1   Useful results

Recall that we write the cumulative distribution function for the standard Gaussian distribution $\Phi(x) = \frac{1}{\sqrt{2\pi}} \int_{-\infty}^{x} e^{-u^2/2} du$. We state the Gaussian isoperimetric inequality [1, 2], the main technical tool used in to prove the results in this paper.

**Theorem A.1** (Gaussian isoperimetric inequality). *Let $\nu_d$ be the Gaussian measure on $\mathbb{R}^d$. Let $A \subseteq \mathbb{R}^d$ and let $A_\eta = \{z \in \mathbb{R}^d : \exists z' \in A \text{ s.t. } \|z - z'\|_2 \leq \eta\}$. If $\nu_d(A) = \Phi(a)$ then $\nu_d(A_\eta) \geq \Phi(a + \eta)$.*

We then state some useful bounds on the cumulative distribution function for the Gaussian distribution $\Phi$.

**Lemma A.1** (see e.g., [3]). *We have for $x \geq 0$,*

$$1 - \frac{e^{-x^2/2}}{\sqrt{2\pi}} \frac{2}{x + \sqrt{x^2 + 8/\pi}} \leq \Phi(x) \leq 1 - \frac{e^{-x^2/2}}{\sqrt{2\pi}} \frac{2}{x + \sqrt{x^2 + 4}} \ . \tag{A.1}$$

**Lemma A.2.** *Let $p \in [1/2, 1]$, we have for all $\eta > 0$,*

$$\Phi(\Phi^{-1}(p) + \eta) \geq 1 - (1 - p)\sqrt{\frac{\pi}{2}} e^{-\eta^2/2} e^{-\eta \Phi^{-1}(p)} \ . \tag{A.2}$$

*If $p = 1 - \frac{1}{K}$ for $K \geq 5$ and $\eta \geq 1$, we have*

$$\Phi(\Phi^{-1}(1 - \frac{1}{K}) + \eta) \geq 1 - \frac{1}{K} \sqrt{\frac{\pi}{2}} e^{-\eta^2/2} e^{-\eta \sqrt{\log\left(\frac{K^2}{4\pi \log(K)}\right)}} \ . \tag{A.3}$$

*Proof.* As $p \geq 1/2$, we have $\Phi^{-1}(p) \geq 0$. Thus,

$$\Phi(\Phi^{-1}(p) + \eta) \geq 1 - \frac{1}{\sqrt{2\pi}} \frac{2e^{-(\Phi^{-1}(p)+\eta)^2/2}}{\Phi^{-1}(p) + \eta + \sqrt{(\Phi^{-1}(p) + \eta)^2 + 8/\pi}}$$

$$= 1 - \frac{1}{\sqrt{2\pi}} \frac{2e^{-\Phi^{-1}(p)^2/2 - \eta^2/2 - \eta \Phi^{-1}(p)}}{\Phi^{-1}(p) + \eta + \sqrt{(\Phi^{-1}(p) + \eta)^2 + 8/\pi}}$$

$$= 1 - \left( \frac{1}{\sqrt{2\pi}} \frac{2e^{-\Phi^{-1}(p)^2/2}}{\Phi^{-1}(p) + \sqrt{\Phi^{-1}(p)^2 + 4}} \right)$$

$$\times \frac{\Phi^{-1}(p) + \sqrt{\Phi^{-1}(p)^2 + 4}}{\Phi^{-1}(p) + \eta + \sqrt{(\Phi^{-1}(p) + \eta)^2 + 8/\pi}} e^{-\eta^2/2 - \eta \Phi^{-1}(p)} \ .$$

Now we use the fact that

$$\left( \frac{1}{\sqrt{2\pi}} \frac{2e^{-\Phi^{-1}(p)^2/2}}{\Phi^{-1}(p) + \sqrt{\Phi^{-1}(p)^2 + 4}} \right) \leq 1 - \Phi(\Phi^{-1}(p)) = 1 - p \, .$$

As a result,

$$\Phi(\Phi^{-1}(p) + \eta)$$

$$\geq 1 - (1-p)e^{-\eta^2/2 - \eta\Phi^{-1}(p)} \frac{\Phi^{-1}(p) + \sqrt{\Phi^{-1}(p)^2 + 4}}{\Phi^{-1}(p) + \eta + \sqrt{(\Phi^{-1}(p) + \eta)^2 + 8/\pi}}$$

$$\geq 1 - (1-p)e^{-\eta^2/2 - \eta\Phi^{-1}(p)} \frac{\Phi^{-1}(p) + \sqrt{\Phi^{-1}(p)^2 + 4}}{\Phi^{-1}(p) + \sqrt{\Phi^{-1}(p)^2 + 8/\pi}}$$

$$\geq 1 - (1-p)e^{-\eta^2/2} e^{-\eta\Phi^{-1}(p)} \frac{\sqrt{4}}{\sqrt{8/\pi}} \, .$$

In the case $p = 1 - \frac{1}{K}$, it suffices to show that that for $K \geq 5$, we have

$$\Phi^{-1}(1 - 1/K) \geq \sqrt{\log\left( \frac{K^2}{4\pi \log(K)} \right)} \, . \tag{A.4}$$

Using the upper bound in (A.1), and the fact that $x + \sqrt{x^2 + 2} \leq 2\sqrt{x^2 + 1}$, it suffices to show that $\frac{1}{2} \frac{e^{-x^2}}{\sqrt{\pi}\sqrt{x^2+1}} \geq \frac{1}{K}$ where $x = \sqrt{\frac{1}{2} \log\left( \frac{K^2}{4\pi \log(K)} \right)}$. This inequality is equivalent to showing that $\sqrt{\log(K)} \geq \sqrt{x^2 + 1}$ for the same value of $x$. If we let $u = \log(K)$ this amounts to showing that $\sqrt{u} \geq \sqrt{u - \frac{1}{2}\log(4\pi u) + 1}$ for all $u \geq \log(5)$. For such $u$ one can verify that $-\frac{1}{2}\log(4\pi u) + 1 \leq 0$ and so clearly the inequality is satisfied.

$\square$

## A.2   Proof of Theorem 1

*Proof.* To prove the general bound in Eq. (2), we define

$$C_{i\rightarrow} = \{x \in C_i : \mathrm{dist}(x, \cup_{j\neq i}C_j) \leq \eta\}.$$

Here, $\mathrm{dist}(x, C)$ is defined as $\inf_{x'\in C} \|x - x'\|$. Let us also introduce the following sets in the $z$-space: $B_i = g^{-1}(C_i)$ and $B_{i\rightarrow} = \{z \in B_i : \mathrm{dist}(z, \cup_{j\neq i}B_j) \leq \omega^{-1}(\eta)\}$. It is easy to verify that $g(B_{i\rightarrow}) \subseteq C_{i\rightarrow}$. Thus we have $\mathbb{P}(C_{i\rightarrow}) = \nu(g^{-1}(C_{i\rightarrow})) \geq \nu(B_{i\rightarrow})$. Now note that $B_{i\rightarrow} \bigcup \cup_{j\neq i}B_j$ is nothing but the set of points that are at distance at most $\omega^{-1}(\eta)$ from $\cup_{j\neq i}B_j$. As such, by the Gaussian isoperimetric inequality (Theorem A.1) applied with $A = \cup_{j\neq i}B_j$ and $a = a_{\neq i}$, we have $\nu(B_{i\rightarrow}(\eta)) + \nu(\cup_{j\neq i}B_j) \geq \Phi(a_{\neq i} + \omega^{-1}(\eta))$, i.e., $\nu(B_{i\rightarrow}) \geq \Phi(a_{\neq i} + \omega^{-1}(\eta)) - \Phi(a_{\neq i})$. As $B_{i\rightarrow}$ are disjoint for different $i$, we have

$$\nu(\cup_i B_{i\rightarrow}(\eta)) \geq \sum_{i=1}^{K} \left( \Phi(a_{\neq i} + \omega^{-1}(\eta)) - \Phi(a_{\neq i}) \right) \, .$$

The proof of inequality (2) of the main text then follows by using $\mathbb{P}(C_{i\rightarrow}) \geq \nu(B_{i\rightarrow})$.

To prove inequality (3), observe that if $\mathbb{P}(C_i) \leq \frac{1}{2}$ for all $i$, then $\mathbb{P}(\cup_{j \neq i} C_j) \geq \frac{1}{2}$ for all $i$. Then we use the bound (A.2) to get,

$$
\begin{aligned}
\mathbb{P}(\cup_i C_{i \to}(\eta)) &\geq \sum_{i=1}^{K} (\Phi(\Phi^{-1}(\mathbb{P}(\cup_{j \neq i} C_j)) + \eta) - \mathbb{P}(\cup_{j \neq i} C_j)) \\
&\geq \sum_{i=1}^{K} (1 - (1 - \mathbb{P}(\cup_{j \neq i} C_j)) \sqrt{\frac{\pi}{2}} e^{-\eta^2/2} - \mathbb{P}(\cup_{j \neq i} C_j)) \\
&= (1 - \sqrt{\frac{\pi}{2}} e^{-\eta^2/2}) \sum_{i=1}^{K} (1 - \mathbb{P}(\cup_{j \neq i} C_j)) \\
&= 1 - \sqrt{\frac{\pi}{2}} e^{-\eta^2/2} \,.
\end{aligned}
$$

For the bound (4) that makes explicit the dependence on the number of classes, we simply use the more explicit bound in (A.3). $\qquad\square$

### A.3 Proof of Theorem 2

*Proof.* Let $x = g(z) \in \mathcal{X}$ and $x' \in \mathcal{X}$. Let $z^*$ be such that $\tilde{f}(x') = f(g(z^*))$. By definition of $\tilde{f}$, we have $\|x' - g(z^*)\| \leq \|x' - g(z)\|$. As such, using the triangle inequality, we get

$$
\begin{aligned}
\|g(z) - g(z^*)\| &\leq \|g(z) - x'\| + \|x' - g(z^*)\| \\
&\leq 2\|g(z) - x'\| \,.
\end{aligned}
$$

Taking the minimum over all $x'$ such that $\tilde{f}(x) \neq \tilde{f}(x')$, we obtain

$$
r_{\text{in}}(x) \leq 2 r_{\text{unc}}(x).
$$

$\qquad\square$

### A.4 Proof of Theorem 3

*Proof.* We use the same notations as in the proof of Theorem 1: let $B_i(f) = g^{-1}(C_i(f))$ and $B_i(h) = g^{-1}(C_i(h))$, and let

$$
B_{i \to} = \{z \in B_i(f) \cup B_i(h) : \text{dist}(x, \overline{B_i(f)} \cap \overline{B_i(h)}) \leq \omega^{-1}(\eta)\}.
$$

where the notation $\overline{B}$ stands for the complement of $B$.

Note that $B_i(f) \cup B_i(h) = \overline{\overline{B_i(f)} \cap \overline{B_i(h)}}$. We have $\nu(\overline{B_i(f)} \cap \overline{B_i(h)}) \geq \nu(\overline{B_i(f)}) - \delta = 1 - \nu(B_i(f)) - \delta \geq \frac{1}{2}$. Thus, using the Gaussian isoperimetric inequality with $A = \overline{B_i(f)} \cap \overline{B_i(h)}$, we obtain

$$
\nu(B_{i \to}) + \nu(\overline{B_i(f)} \cap \overline{B_i(h)}) \geq 1 - \left(1 - \nu(\overline{B_i(f)} \cap \overline{B_i(h)})\right) \sqrt{\frac{\pi}{2}} e^{-\eta^2/2},
$$

where we also used inequality (A.2). As a result,

$$
\begin{aligned}
\nu(B_{i \to}) &\geq (1 - \nu(\overline{B_i(f)} \cap \overline{B_i(h)}))(1 - \sqrt{\frac{\pi}{2}} e^{-\eta^2/2}) \\
&\geq \nu(B_i(f))(1 - \sqrt{\frac{\pi}{2}} e^{-\eta^2/2}) \,.
\end{aligned}
$$

Now assume that $z \in B_{i \to}$ but also $z \in B_i(f) \cap B_i(h)$. Then it is classified as $i$ for both $f$ and $h$. In addition, the condition $z \in B_{i \to}$ ensures that there exists $z' \in \overline{B_i(f)} \cap \overline{B_i(h)}$ such that $\|z - z'\|_2 \leq \omega^{-1}(\eta)$. Setting $v = g(z') - g(z)$, we have that $f(g(z) + v) \neq f(g(z))$ and $h(g(z) + v) \neq h(g(z))$ and $\|v\| \leq \omega(\|z - z'\|) \leq \eta$. As such it suffices to show that the set $B_{i \to} \cap (B_i(f) \cap B_i(h))$ has sufficiently large measure. Indeed, we have

$$
\begin{aligned}
&\nu(B_{i \to} \cap (B_i(f) \cap B_i(h))) \\
&\geq \nu(B_{i \to}) - \nu(B_i(f) \cap \overline{B_i(h)}) - \nu(\overline{B_i(f)} \cap B_i(h)) \,.
\end{aligned}
$$

Summing over $i$, we get

$$\sum_{i=1}^{K} \nu(B_{i\to} \cap (B_i(f) \cap B_i(h))) \geq 1 - \sqrt{\frac{\pi}{2}} e^{-\eta^2/2} - 2\delta \,,$$

because $\sum_{i=1}^{K} \nu(B_i(f) \cap \overline{B_i(h)}) + \nu(\overline{B_i(f)} \cap B_i(h)) = 2 \cdot \nu \{f \circ g(z) \neq h \circ g(z)\} \leq 2\delta$.

$\square$

## A.5 Proof of Theorem 4

*Proof.* We first treat the case $\delta = 0$. Given $z$ we denote by $r_{\mathcal{Z}}(z) = \min \{\|r\|_2 : f(g(z+r)) \neq f(g(z))\}$. Then it is easy to see that $r_{\text{in}}(g(z)) \leq \omega(r_{\mathcal{Z}}(z))$. As such we have $\mathbb{E}_x[r_{\text{in}}(x)] = \mathbb{E}_z[r_{\text{in}}(g(z))] \leq \mathbb{E}_z[\omega(r_{\mathcal{Z}}(z))] \leq \omega(\mathbb{E}_z[r_{\mathcal{Z}}(z)])$. Now we have

$$\mathbb{E}_z[r_{\mathcal{Z}}(z)] = \int_0^\infty \mathbb{P}_z[r_{\mathcal{Z}}(z) \geq \eta] d\eta.$$

Using a bound similar to Theorem 1 applied to $r_{\mathcal{Z}}$ we get

$$\mathbb{E}_z[r_{\mathcal{Z}}(z)] \leq \int_0^\infty \left(1 - \sum_{i=1}^{K} \Phi(a_{\neq i} + \eta) - \Phi(a_{\neq i})\right) d\eta$$

$$= \sum_{i=1}^{K} \int_0^\infty \Phi(-a_{\neq i} - \eta) d\eta$$

where in the equality, we used the fact that $1 = \sum_{i=1}^{K}(1 - \Phi(a_{\neq i}))$. Now observe that for any $a \in \mathbb{R}$,

$$\int_0^\infty \Phi(-a - \eta) d\eta = \int_a^\infty \int_{-\infty}^{-u} \frac{e^{-t^2/2}}{\sqrt{2\pi}} dt du$$

$$= \int_{-\infty}^\infty \left(\int_a^\infty \mathbf{1}_{t \leq -u} du\right) \frac{e^{-t^2/2}}{\sqrt{2\pi}} dt$$

$$= \int_{-\infty}^\infty (-t - a) \mathbf{1}_{a \leq -t} \frac{e^{-t^2/2}}{\sqrt{2\pi}} dt$$

$$= \frac{e^{-a^2/2}}{\sqrt{2\pi}} - a\Phi(-a).$$

As a result,

$$\mathbb{E}_z[r_{\mathcal{Z}}(z)] \leq \sum_{i=1}^{K} -a_{\neq i} \Phi(-a_{\neq i}) + \frac{e^{-a_{\neq i}^2/2}}{\sqrt{2\pi}}$$

This establishes the first inequality.

Assuming now that the classes are equiprobable, i.e., $a_{\neq i} = \Phi^{-1}(1 - 1/K) =: a(K)$ for all $i$ we get that

$$\mathbb{E}[r_{\text{in}}(x)] \leq \omega\left(-a(K)^2 + \frac{K}{\sqrt{2\pi}} e^{-a(K)^2/2}\right).$$

Using the bound (A.4) on $a(K)$ we get:

$$\mathbb{E}[r_{\text{in}}(x)] \leq \omega\left(\sqrt{2\log(K)} - \sqrt{2\log(K) - \log(4\pi \log(K))}\right)$$

$$= \omega\left(\frac{\log(4\pi \log(K))}{\sqrt{2\log(K)} + \sqrt{2\log(K) - \log(4\pi \log(K))}}\right)$$

$$\leq \omega\left(\frac{\log(4\pi \log(K))}{\sqrt{2\log(K)}}\right)$$

Figure 1: Left: Illustration of checkerboard example. Right: Lower bound on robustness as a function of $\eta$ for the general result in Theorem 1 (blue curve) and the checkerboard example in Eq. A.5 (red curve).

We assume now that $g$ is such that $W(g_*(\nu), \mu) \leq \delta$, where $W$ denotes the Wasserstein distance in $(\mathcal{X}, \|\cdot\|)$. Let $(X, X')$ be a coupling with $X \sim \mu$ and $X' \sim g_*(\nu)$. We will construct a random variable $X''$ such that almost surely $X''$ and $X$ are classified differently. We define $X'' = X'$ if $X$ and $X'$ are classified differently and otherwise $X'' = X' + \vec{r}^*(X')$ where $\vec{r}^*(X')$ is defined to be a vector of minimum norm such that $X' + \vec{r}^*(X')$ and $X'$ are classified differently. Then we have

$$\underset{x \sim \mu}{\mathbb{E}} \, r_{\text{unc}}(x)$$
$$\leq \mathbb{E}\|X - X''\|$$
$$= \mathbb{E}(\mathbf{1}_{f(X) \neq f(X')} \|X - X'\|) + \mathbb{E}(\mathbf{1}_{f(X) = f(X')} \|X - (X' + \vec{r}^*(X'))\|)$$
$$\leq \mathbb{E}\|X - X'\| + \mathbb{E}\|\vec{r}^*(X'))\| \,.$$

By choosing a coupling such that $W(g_*(\nu), \nu) = \mathbb{E}\|X - X'\|$, we get $\mathbb{E}\|X - X'\| \leq \delta$. In addition, $\mathbb{E}\|\vec{r}^*(X'))\| \leq \mathbb{E}_x r_{\text{in}}(x)$. The statement therefore follows. $\qquad \square$

## B Toy example: tightness of Theorem 1

As an illustration to Remark 4, we explicitly show through a toy example that a classifier which is not linear in the $\mathcal{Z}$-space can be significantly less robust than a linear one.

**Example B.1** (Checkerboard class partitions). *Assume that $B_1 = g^{-1}(C_1)$ and $B_2 = g^{-1}(C_2)$ are given by:*

- $B_1 = \{(z_1, \ldots, z_d) : \sum_{i=1}^{d} \lfloor z_i \rfloor \mod 2 = 0\}$,

- $B_2 = \mathbb{R}^d - B_1$.

*See Fig. 1a for an illustration. Then, we have*

$$\mathbb{P}(z \in B_1 \text{ and } \text{dist}(z, B_2) \leq \eta) + \mathbb{P}(z \in B_2 \text{ and } \text{dist}(z, B_1) \leq \eta) \geq 1 - (1 - \eta)^d. \quad (A.5)$$

*Fig 1b compares the general bound in Theorem 1 to Eq. (A.5). As can be seen, in the checkerboard partition example, the probability of fooling converges much quicker to 1 (wrt $\eta$) than the general result in Theorem 1. Hence, a classifier that creates many disconnected classification regions can be much more vulnerable to perturbations than a linear classifier in the latent space.*

*Proof.* We have $\nu(B_1) = \nu(B_2) = \frac{1}{2}$. Let $z \in \mathbb{R}^d$ in $B_2$ be such that for some $i \in \{1, \ldots, d\}$, $z_i - \lfloor z_i \rfloor \in [0, \eta) \cup (1 - \eta, 1)$, then $z - \eta e_i \in B_1$ or $z + \eta e_i \in B_1$, and thus $z$ is at distance at most $\eta$

from $B_1$. As a result, if $z$ is at distance $> \eta$ from $B_1$, then for all $i \in \{1, \dots, d\}$, $z_i - \lfloor z_i \rfloor \in [\eta, 1 - \eta]$. As a result,

$$
\begin{aligned}
&\mathbb{P}_z(z \in B_2, \text{dist}(z, B_1) > \eta) \\
&\leq \mathbb{P}_z(z \in B_2, \forall i, z_i - \lfloor z_i \rfloor \in [\eta, 1 - \eta]) \\
&= \frac{1}{\sqrt{2\pi}^d} \sum_{\substack{(j_1, \dots, j_d) \in \mathbb{Z}^d, \\ j_1 + \dots + j_d \mod 2 = 1}} \int_{j_1 + \eta}^{j_1 + 1 - \eta} dz_1 \cdots \int_{j_d + \eta}^{j_d + 1 - \eta} dz_d e^{-\frac{\sum_i z_i^2}{2}} .
\end{aligned}
$$

Now observe that for any $j \in \mathbb{Z}$, as the function $z \mapsto e^{-z^2/2}$ is monotone on the interval $[j, j+1]$ (nondecreasing if $j < 0$ and nonincreasing if $j \geq 0$). Thus, we have $\int_{j+\eta}^{j+1-\eta} e^{-\frac{z^2}{2}} dz \leq (1 - \eta) \int_{j}^{j+1} e^{-\frac{z^2}{2}} dz$, when $\eta \leq \frac{1}{2}$. As a result,

$$
\begin{aligned}
&\mathbb{P}_z(z \in B_2, \text{dist}(z, B_1) > \eta) \\
&\leq \frac{1}{\sqrt{2\pi}^d} (1 - \eta)^d \sum_{\substack{(j_1, \dots, j_d) \in \mathbb{Z}^d, \\ \sum_i j_i \mod 2 = 1}} \int_{j_1}^{j_1 + 1} dz_1 \cdots \int_{j_d}^{j_d + 1} dz_d e^{-\frac{\sum_i z_i^2}{2}} \\
&= (1 - \eta)^d \mathbb{P}_z(z \in B_2) \\
&= \frac{1}{2}(1 - \eta)^d .
\end{aligned}
$$

With the same reasoning, $\mathbb{P}_z(z \in B_1, \text{dist}(z, B_2) > \eta) \leq \frac{1}{2}(1 - \eta)^d$ and gives inequality (A.5). $\quad\square$

## C  Experimental results

### C.1  Details of the used models

For the SVHN dataset, we resize the images to $64 \times 64$. For the generative model, we use the PyTorch implementation of DCGAN available on `https://github.com/pytorch/examples/blob/master/dcgan/main.py` using the default parameters for architecture and optimization. The 2-layer LeNet classifier has the following architecture:

$$
\begin{aligned}
&\text{Conv}(5, 2, 16) \to \text{ReLU} \to \text{MaxPool}(4) \\
&\to \text{Conv}(5, 2, 32) \to \text{ReLU} \to \text{MaxPool}(4) \to \text{FC}(10),
\end{aligned}
$$

where the parameters of Conv are kernel size, padding and number of filters, respectively. We used the ResNet18 and ResNet101 architectures available on `https://github.com/kuangliu/pytorch-cifar/blob/master/models/resnet.py`, with a kernel size of 5 for Conv1 and a stride of 2. For all 3 architectures, we used SGD with a learning rate of 0.01, momentum of 0.9, batch size of 100. To solve the problem in Eq. A.6, we use gradient descent (for the maximization of $\|g(z) - g(z')\|_2$) with learning rate 0.1 for 1,000 steps. The upper bound was computed based on 100 samples of $z$.

For the CIFAR-10 experiment, we use a similar DCGAN generative model. The VGG-type architecture has 11 conv layers, each of kernel size 3, with number of output channels $(64, 64, 128, 128, 128, 256, 256, 256, 512, 512, 512)$ and stride $(1, 1, 2, 1, 1, 2, 1, 1, 2, 1, 1)$. Each conv layer is followed by BatchNorm and a ReLU function. For the WideResNet architecture, we use the WRN-28-10 model available on `https://github.com/szagoruyko/wide-residual-networks`. SGD is used with learning rate 0.1, momentum 0.9, and batchsize 100. For the adversarially trained Wide ResNet with PGD training, we have used the model of [4].

### C.2  Numerical evaluation of the upper bound

To evaluate numerically the upper bound, we have used a probabilistic version of the modulus of continuity, where the property is not required to be satisfied for *all* $z, z'$, but rather with high probability, and accounted for the error probability in the bound. Specifically, while the modulus

of continuity function is given by $\omega(\delta) = \max_z \max_{z':\|z-z'\|_2 \leq \delta} \|g(z) - g(z')\|_2$, we use in the experiments a probabilistic version of the modulus of continuity, given by:

$$\omega_\kappa(\delta) = \min \left\{ \alpha : \mathbb{P} \left( \sup_{z':\|z-z'\|_2 \leq \delta} \|g(z) - g(z')\|_2 \geq \alpha \right) \leq \kappa \right\}. \tag{A.6}$$

Then, the following bound holds for any $\delta, \kappa$:

$$\mathbb{P}\left(r_{\text{in}}(x) \geq \omega_\kappa(\delta)\right) \leq \kappa + \underbrace{\mathbb{P}\left(\exists r : \|r\|_2 \geq \delta : f(g(z+r)) \neq f(g(z))\right)}_{1- \text{ probability in Theorem 1 with } \omega \text{ identity.}}. \tag{A.7}$$

For example, when $\kappa$ is set to $0$, we recover the exact bounds in Theorem 1. When $\kappa > 0$, we have to account for the use of a probabilistic definition of the modulus of continuity in the bound; this exactly corresponds to the additive $\kappa$ term in the probability in Eq. (A.7).

In practice, for a fixed target probability (set to $0.25$ in the experiments of the main paper), it is possible to choose the value of $\delta$ that yields the best bound, since Eq. (A.7) is valid for any $\delta$. For a fixed value of $\delta$, we used gradient descent (until the loss function stabilizes) in order to solve the optimization problem $\sup_{z:\|z'-z\|_2 \leq \delta} \|g(z) - g(z')\|$. For a fixed value of $\delta$, we hence summarize the procedure used to evaluate the upper bound in Algorithm 1. We have used in practice 100 samples to estimate the upper bound, for each value of $\delta$. For any value of $\delta$, Algorithm 1 provides an estimate of the upper bound; such an estimate can be improved by using many different values of $\delta$.

---

**Algorithm 1** Numerical evaluation of the upper bound.

---
1: // **input**: $\delta$, target probability $p_t$.
2: // **output**: numerical upper bound.
3: $p \leftarrow p_t - p_u(\delta)$. // $p_u(\delta)$ is the probability from Theorem 1 with $\omega$ set to identity.
4: **repeat**: $i = 1, \ldots$
5:     Sample $z_i \sim \mathcal{N}(0, I_d)$.
6:     Compute $s_i \leftarrow \sup_{z':\|z_i-z'\|_2 \leq \delta} \|g(z_i) - g(z')\|$.
7: **until** enough samples are taken
8: Use the above $s_i$ to estimate $\alpha$ such that $\tilde{\mathbb{P}}(s_i \geq \alpha) \leq p$, where $\tilde{\mathbb{P}}$ is the empirical probability distribution.
    **return** $\alpha$.

---

### C.3 Illustration of generated images

Fig. 2 illustrates generated images for SVHN, as well as corresponding perturbed images that fool a ResNet-18 classifier (*in-distribution* robustness). Similarly, Fig. 3 illustrates examples of generated images for CIFAR-10, as well as perturbed samples required to fool the VGG classifier, where perturbed images are constrained to belong to the data distribution (i.e., *in-distribution* setting).

Figure 2: Examples of generated images with DCGAN for the SVHN dataset, and associated perturbed images (*in-distribution* perturbations). For each pair of images, the left shows the original image, and the right shows the perturbed image. The estimated label (using ResNet-18) of each image is shown on top of each image.

## Footnotes

*Univ Lyon, ENS de Lyon, CNRS, UCBL, LIP, F-69342, Lyon Cedex 07, France