[Reviews · NeurIPS 2018]

Reviewer 1



In this paper, the authors study the robustness of a classifier with respect to perturbations in the input. In particular, the authors obtain high probability upper bounds on robustness with respect to perturbations within the input manifold in terms of the norm of perturbation. The input manifold is modeled using a network g that maps from a latent space to the input space. The authors further show that for any classifier with in-distribution robustness r, one can obtain a classifier with unconstrained robustness r/2. The results are generalized to the scenario when the network g does not model the input manifold perfectly. The paper is very well written and easy to read. A theoretical understanding of how classifiers behave when the input is perturbed is necessary in building robust classifiers. In this regard, the work done in this paper appears significant. In particular, the result that maximal robustness with respect to perturbations occur when the induced classification boundaries in the latent space are linear, can be used to construct robust classifiers.

Reviewer 2



This paper introduces several results regarding the robustness of classifiers in terms of misclassification under adversarial examples. Providing a straightforward definition of robustness, the work shows that there exists an upper bound on robustness for any classifier, which is worsened by the dimensionality of the latent space or the classification task. This upper bound can be attained, achieving maximal robustness, by having classifiers inducing linearly separable regions in the latent space. Systematic modification of classifiers is also proposed to achieve unconstrained robustness close to in-distribution robustness. These results are supported by theoretical analysis and some empirical studies. The work brings some enlightening results involving the robustness of classifiers under adversarial examples. The paper is well written, the results are novel, and their exposition is of high quality. I propose to accept this paper as it would contribute to the community significantly. I consider the work to be showing a series of negative results diminishing the current optimism around generative models and bringing some realism to such endeavors. As the theory suggests some ways of achieving the upper bound, it would have been encouraging to show some empirical demonstration of that. For example, it might be shown that a state-of-the-art method is severely affected by adversarial examples on a real dataset due to not achieving the upper bound of robustness, but a modification to the classification as suggested by the work alleviates this issue. Additionally, it would be useful to know if there exists an example on a real dataset where the upper bound of robustness is highly reduced due to a large number of classes and getting around this would be hopeless. Or, is it the case that we can always be hopeful of getting around it by low-dimensional latent space with non-smooth generative models? *** After author rebuttal *** Excellent paper and excellent replies. Thanks!

Reviewer 3



Description The work theoretically studies the question of robustness of classifiers (in particular NNs) to adversarial attacks. The main result is an upper bound on such robustness of classifiers derived under certain assumptions. It is very general in the sense that the upper bound applies to all classifiers not necessarily NNs. The bound depends on the classification boundaries (in particular the number of classes) and on how “smooth” the distribution of data (images to classify) is. The bound can be computed numerically when the generative model is known. Authors propose experiments with DCGAN generated images and several classification networks architectures. The bound appears rather tight / useful in these experiments. There is an interesting discussion of the consequences. Evaluation I thank the authors for their work. It is of high quality, and mostly clear and technically accurate. Below I question some of the assumptions / conclusions, which determines the degree of relevance of the work to the practical problems and phenomena of (in)stability of NNs. Specifically, 1) classifying all points of a smooth generative model may be an inappropriate task formulation and 2) conclusions about the dependence on the dimensionality d are unclear or possibly incorrect. My current impression is that the general bound and further results: the relation between in-distribution and unconstrained robustness and transferability are derived under the definition of robustness which is not fully practically relevant. I am not certain about utility of these results, but the paper is definitely a significant contribution and a worthy publication. Questions Q1. The work starts with the motivation that NNs are vulnerable to adversarial attacks. The work also arrives at the conclusion that robustness of any classifier is rather limited, which correlates with the above motivation but may be unrelated to it. The derived theory is not specific to NNs, and thus cannot help to find out what is wrong with NNs specifically. The fact that a significant portion of data points admit small norm perturbations is coming from the combination of assumptions on smoothness and that we wish to classify the whole distribution, which I think is not realistic. Let me give an example: consider a generative model of human faces. A convex combination of latent variables corresponding to faces of two existing persons would generate a likely face (even more probable due to the normal distribution prior). It can be easily misclassified as either of the two persons. The paper builds on the fact that the fraction of such points that are likely to be generated and are ambiguous to classify is relatively large. The assumptions are thus not fully realistic together: either we need to consider a “real” face distribution of existing people (not smooth) or request the classification of the relevant subset only. I think the obtained result is unrelated to vulnerability of NNs, in particular they are tested with real distributions. Q2. The dependence on dimensionality d is unclear. In the related work section 2, d denotes the dimensionality o the data space (=images). In the formal part, it is the dimensionality of the latent space. It is specifically unclear at line 124: “||x|| scales as \sqrt(d)$. I do not see why this would be true for $d$ being latent dimension. On the other hand: line 124: “assuming L is constant” is unclear. For example, if one increases the data dimension by upsampling the generated images, $L$ grows proportionally. In another place the paper mentions that the input data is considered to be normalized (thus ||x|| would have to be constant). It seems that all these assumptions cannot be used simultaneously. Please clarify. Discussion It seems to me that the work is related to classical results establishing upper bounds on the classification accuracy. The reason for a general bound would be the same: the confusion of the data distribution. The bound is relevant when this becomes a prevailing problem (I believe this is the case in the experiments). Otherwise, other sources of classification errors / instability prevail. Minor: The idea of stability “under generative model” was not clear until section 3 to me, maybe could be explained earlier. Final remarks I still have the concern of relevance. The authors argue that existing generators are producing images “from the training distribution”. This does not convince me since the prior distribution is Gaussian and therefore all latent states in the convex hull of latent representations of the training data are likely. They will unavoidably generate images that are inbetween two or more classes, confusing even for human. For example a face between man and woman. Small perturbations easily move them over the imaginary classification boundary. Imaginary is because these intermediate faces are not in the real data distribution. As another example consider the classification by a feature with overlapping conditional Gaussian distributions. Even the best Bayesian classifier will have a non-zero error. The submission shows a bound on stability related to such data overlap. This is different from sensitivity of NNs to small input perturbations in the case when training set is disconnected, consisting of several clusters. Furthermore, the paper studies experimentally networks trains for generation only. A more appropriate case for study would be a generator-discriminator pair such as semi-supervised variational autoencoders